# Urban Living Labs and Critical Infrastructure Resilience: A Global Match?

Erick Elysio Reis Amorim [1,2,*] , Monique Menezes [2,3] and Karoline Vitória Gonçalves Fernandes [2]

1    Brazilian School of Public and Business Administration, Getúlio Vargas Foundation,
     Rio de Janeiro 22231-010, RJ, Brazil
2    Center for Efficiency in Urban Sustainability, Federal University of Piauí, Teresina 64048-500, PI, Brazil
3    Department of Political Science, Federal University of Piauí, Teresina 64048-500, PI, Brazil
*    Correspondence: erickelysio@gmail.com

**Abstract:** The challenges to public policy brought by climate change are some of the biggest challenges for cities around the world. These challenges are costlier and more substantial for low-income communities given the existence of their greater social and economic vulnerability. Among the existing tools, this paper highlights the role played by urban living labs (ULLs), which have been discussed in the literature as a booster of urban resilience in a more sustainable direction. By considering ULLs as strategic institutional arrangements that seek resilience for the critical urban infrastructure challenges of climate change, the main target of this paper is to analyze ULLs as a strategy for increasing critical infrastructure resilience in the region of the Global South. These labs were initiated in developed countries, so we can ask: How are developing countries adapting this strategy in order to mitigate the problems of climate change? To achieve this goal, we reviewed previous literature on ULLs, specifically looking for case studies with ULL projects and highlighting the processes of public innovation policies and transfers of knowledge between countries; in order to complement our empirical analysis, we carried out a case study on Brazil. Despite the limitations of the sample, the data suggest the existence of different barriers to the implementation of ULL projects in Brazil compared to those in European cities.

**Keywords:** cities; urban living labs; critical infrastructure; innovation; climate change

## 1. Introduction

Although cities occupy only 3% of the Earth's surface, urban areas are home to 55% of the world's population, and this number is expected to continue growing to 68 percent by 2050 [1]. As cities grow, so does the energy consumption and the pressure on natural resources. However, as Glaeser [2] argues, urban life is better for humanity economically, socially, and, most of all, in terms of environment, since a compact city has scale economies that result in lower energy consumption and greenhouse emissions. As centers of development, the future of global sustainability will be determined by how well cities implement mitigation and adaptation measures [3]. By recognizing this dichotomy between vulnerability and opportunity, cities worldwide are becoming protagonists in climate governance, and reports of cities aiming to reach net-zero gas emission targets by 2050 or before are increasingly heard in global climate discourse [4].

Responding to climate change is one of the central challenges of our time. Low-income countries and communities are generally more vulnerable to the effects of climate change and will also bear significant costs during the transition to low-carbon economies [5]. The approach to climate change requires that people and societies not only overcome the complex economic, political, technological, and social challenges, but also circumvent various cognitive biases and prejudices [6]. Financing social infrastructure is one of the biggest challenges to promoting more sustainable urbanization. However, over the last

decade, 80–85% of all infrastructure investments in developing countries have been funded by the public sector [7].

Cities are adopting mechanisms for experimentation and innovation as possible sources of solutions to new and old urban problems. Within this context, so-called "smart cities" have been studied and implemented over recent decades [8,9]. While smart cities' concept focused initially on using "smart technologies" to address social, economic, and environmental problems more efficiently, the focus is slowly shifting towards a citizen-centric approach called "smart governance" [10]. The idea of smart cities was widely spread across the world during the 1990s, as it was one of the predecessor concepts to what we know today as the "living lab" [11,12]. Nowadays, we have seen the expansion of the urban living lab (ULL), which can be placed in the fertile ground of urban experimentation and active citizen involvement in the creation of innovative solutions to the main challenges faced by contemporary cities. It is important to study ULLs as some of the most promising real-world laboratory methodologies (RwLs), along with other experimental research approaches, such as urban labs, city labs, community-based initiatives, and niche experiments in social innovation [13,14]. The subject of ULLs aligns with New Public Management in the pursuit of new forms of local partnership and system innovation [15].

The main target of this paper is to analyze ULLs by considering them as strategic institutional arrangements that seek resilience for critical urban infrastructure resilience in the face of the challenges of climate change. As this is a movement that started in developed countries, our article seeks to understand how developing countries can adapt and adopt this strategy to mitigate climate change problems. The literature [16] points out the arise of these laboratories in Europe and the USA. However, we have also observed the creation of ULLs in countries in the Global South and identified a gap in the literature about the management aspects of ULLs. Therefore, our argument is that the development of ULLs in developing countries may follow a different logic from that found in the countries of the Global North countries because of either the scarcity of resources or the greater complexity of public problems.

To offer some enlightenment, a mixed methodological approach that involved a critical analysis of the literature on the topic and a descriptive case study of Brazil was used. Furthermore, this paper discusses the process of production of public innovation policies, resulting in a critical analysis of the transposition of public policy tools from the context of developed countries to the Global South. Our empirical findings show that ULLs from Brazil share similarities with those in Europe, but they face different barriers. It is worth emphasizing that these results refer specifically to only one country in the Global South, Brazil, and that they are those of a preliminary study. However, the findings provide insights into a new research agenda on ULLs in other low-income countries and tools for practitioners that decide on ways to implement these labs in countries in the Global South.

The paper is structured into six sections, including this introduction and a conclusion. The second section discusses the methodological strategies used to achieve the objectives of this work in more detail. Then, key concepts for understanding ULLs—innovation in public policy and critical infrastructure—are addressed. After that, the results of a literature review and a verification of the themes and cities with ULL projects are presented, followed by a discussion on sharing experiences and possible adaptations for the Global South. Next, some experiences of self-declared urban living laboratories in Brazil are presented based on a qualitative and limited sample, with a descriptive analysis of the objectives of these laboratories. Finally, discussions, limitations, concluding remarks, and a suggested research agenda are provided.

## 2. Materials and Methods

As mentioned in the introduction, our research uses a mixed methodological approach in which we combine qualitative and quantitative methods in a single study [17]. A mixed method allows triangulation between qualitative and quantitative data, enabling researchers to use the benefits of both methodological approaches. This methodological

choice was due to the need for different data sources to answer our research question [17]. This need is mainly because this is a topic that has been underexplored in the literature, with many studies on ULLs in developed countries, but few focusing on the Global South.

Along with the concepts and definitions used in the public policy literature, our first strategy was a bibliometric analysis of the literature to investigate the attention given by the literature to many aspects of ULLs. Using the Scopus database, with a systematic approach and practical screening techniques [18,19], we analyzed published sixty-six papers with a focus on two main questions: What is the geographic distribution of the researchers? When case studies were presented, which countries hosted more interventions and in what sector were they related to the critical infrastructure of cities?

Our second strategy was a qualitative survey of some ULLs in Brazil. This choice was made in order for a descriptive and qualitative analysis to be used to get to know some existing ULLs in the country. In this sense, we consider that one of the contributions of this article is the detailed description of some Brazilian ULLs and a comparison with the characteristics of existing laboratories in European countries.

Since the country does not maintain a network of LLs and ULLs, as in Europe, the strategy for locating the ULLs was an active online search. The classification of an innovation laboratory as a ULL followed two criteria: (i) performance in the urban environment; (ii) acting on issues related to sustainability. These criteria were used based on the definition of ULLs presented in the next sections of this paper.

After the identification of these laboratories, we collected information available on websites and sent e-mails with a script. Our focus was on gathering information related to the following dimensions: institutional design, areas of expertise, and interactions between actors. Information was collected from 13 ULLs in Brazil. This was a qualitative and non-representative sample of the existing ULLs in the country. Despite this limitation, this allowed us to carry out some descriptive analyses of these laboratories and provided us with insights into of the models that are being implemented in Brazil.

Conducting such a case study in Brazil is justified for two reasons. First, it is one of the main economies of the Global South, as it is the most important economy in Latin America. Second, although there is a movement to create ULLs in the country, there are few studies on the subject there [12,16]. In this sense, this article can help in understanding this process in Brazil and provide some insights into what is happening in countries with similar characteristics and public challenges.

## 3. Definitions and Conceptualization

In this section, we discuss the concepts of innovation, public policy, and adaption of critical infrastructure to understand the development of ULLs. Considering that ULLs emerged as an attempt to respond to complex public problems in a more sustainable way [20], it is crucial to understand the innovation process in public policies and the importance of innovation in the process of adaptation of critical infrastructure in a more sustainable way.

### 3.1. Critical Infrastructure and Public Innovation Policies

The multidisciplinary character of the public policy field allowed the development of a wide literature on its conceptual definition [21–24]. In this paper, we will adopt the definition by Peters [25], in which public policy is comprised of a set of government activities in which it seeks to influence—directly or by delegation—people's lives. In this sense, public policy can be considered as "government in action" with the objective of implementing policies that seek to solve public problems and, consequently, result in a better quality of life for citizens.

To didactically understand the dynamics of public policies, we use what the literature calls the policy cycle. This analytical model allows us to interpret the public policy in interdependently organized phases and sequences [23]. There is no consensus in the literature about these phases of the public policy cycle; hence, we will adopt the cycle presented by

Secchi [23], in which there are seven steps, namely: identification of the problem, formation of the agenda, formulation of alternatives, decision making, implementation, evaluation, and extinction. For the purposes of this study, we are interested in all of the above, i.e., the public policy production process.

According to Cavalcante et al. [26], the public policy development process has undergone important changes in recent decades with the incorporation of information and communication technologies (ICTs) and the inclusion of civil society in various stages of policy construction, such as hearings and public consultations, participatory budgeting, etc. However, the authors argue that the traditional understanding of the public policy production process is not enough to face increasingly complex public problems.

These problems include, in particular, the challenges of cities in developing countries, in which there is a combination of social vulnerability and the challenges generated by both disorderly growth and climate change. This combination would be an example of what Bentley [27] calls a challenge for formulators and implementers of public policies, given that there is enormous difficulty in presenting clear solutions. These are wicked problems that cross different areas of public policy, professional segments, institutions, and jurisdictions. The model of public policy production dominant in the 20th century, in which there is direct causality between planning, decision, actions and processes, results, and impacts, started to be questioned because it did not obtain satisfactory answers to these wicked problems.

Understanding the context in which individuals are willing to support public policies that aim for the mitigation of climate change has important consequences for the legitimacy, costs, effectiveness, and longevity of implementing any policy alternative [28]. For example, although previous studies provided evidence of the negative effects of CO2 emissions from individual transportation on the environment and human health, these arguments were not enough to change the population's perception of the use of public transport as having a mitigating effect on climate change [29].

The approach to climate change requires that people and societies not only overcome the complex economic, political, technological, and social challenges, but also circumvent various cognitive biases and prejudices. Humans are much more concerned with the present than with the future, and many of the worst impacts of climate change may occur several years from now [5]. Looking forward, and in accordance with the United Nations (UN) 2030 global agenda, where the Sustainable Development Goal (SDG) 11 is "making cities and human settlements including, safe, resilient, and sustainable", it is evident that investment in sustainable urbanization is mandatory.

In this context, innovation in public policies can bring to light this dichotomy between present and future, offering solutions or ideas that do not represent an apparent sacrifice in the present. We hold that innovation is especially important in the area of the adaptation of critical infrastructure given its importance in different areas of citizens' daily lives and its operational costs.

Critical infrastructure is a concept that is in constant transformation. Traditionally, the classification of a certain type of public or private provision as critical has been linked to national security with respect to either internal or external threats. According to the most recent consolidation of 25 countries in the Organization for Economic Co-operation and Development (OECD) [30], the main sectors designated as critical infrastructure by the national governments were energy (25), ICTs (information and communication technologies) (23), transportation (23), health (22), water (22), finance (22), government (17), food supply (17), chemical industry (15), public safety (15), law enforcement (10), nuclear (10), dams and flood defense (7), critical manufacturing (7), the defense industry (5), and the space sector (4). These data reflect the focus of governments on allocating the infrastructure so as to maintain the economic status quo at the expense of the wellbeing of the population.

One of the major threats now incorporated within this debate is climate change, and this aspect is even more relevant to cities [31]. The literature on critical infrastructure is extensive, and creating a single classification is not an easy task. For the purposes of this

paper, we will use the concepts and classifications from the 2022 study by Huddleston and colleagues [32], who carried out a systematic literature review of 86 documents based on Scopus results. The authors highlighted that: (i) Critical infrastructure is linked to the running of a functional society; (ii) the most often mentioned sectors in the literature were energy, transportation, water, and communications; (iii) housing, elderly and early childhood care, legal systems, the construction industry, media, and research were among the sectors with fewer mentions. The types of CI identified, the main characteristics, and examples thereof are summarized in the table below.

Although no common definition of adaptive critical infrastructure was identified, the authors suggested a definition based on the identified types of CI found in the literature (and summarized in Table 1), which we found suitable for our research: Adaptive critical infrastructure comprises tangible and/or intangible systems that are vital for supporting human life and necessary for achieving social, cultural, economic, and environmental outcomes [32] (p. 70). With this broad definition, we will classify the many ULLs found in the literature and their relations to the adaptation of critical infrastructure.

**Table 1.** Types of critical infrastructure.

| Type of CI | Main Characteristics | Examples |
|---|---|---|
| Physical | System that requires large investments, cross-jurisdictional assets, multiple stakeholders/owners, assets with long lifespans, and aging assets. These systems include connections within and between sectors that rely on one another to function properly. | Energy, transportation, water, and communications grids |
| Green or nature-based | Green critical infrastructure captures the network of environmental features—both natural and constructed. It has the focus on actions for protecting, sustainably managing, and restoring natural or modified ecosystems. Normally, there is a focus on off-grid and self-sufficient solutions. | Solar panels, wetlands, and reef restoration |
| Institutional | Vital for ensuring that critical physical infrastructure can achieve its purpose of providing goods and services. Often, the phrase "socio-technical" is used to capture the interplay between physical and human systems. Intangible, deliberately built systems. | Skilled workforce, effective policies, and inter-agency relationships |
| Cultural | Intangible systems that people and communities can create organically. | Culture, connection to a place, sense of community, and local knowledge |

The debate on urban CI is an important subject, since a rapidly urbanizing world devoid of economic opportunities for the majority living in the peripheries of cities worldwide needs responsible and innovative responses to present and future outbreaks from public administrations [33]. Therefore, investment in sustainable urbanization with new technologies and management strategies is mandatory. In the latest report by the Intergovernmental Panel on Climate Change, the chapter on urban systems and other settlements [34] estimated that an investment of over USD 50 trillion is needed to tackle the climate change challenges within cities.

It is important to highlight that innovation does not necessarily mean creation or transformation using technology. In the case of the public sector, innovation consists of the development of a new idea aimed at creating value for society, either by improving the provision of public services to society or by changing internal processes [35]. Thus, innovation in the public sector encompasses four characteristics, namely: novelty in a specific organization or service; the need for dissemination of the invention; the relationship between the process and the product, in the sense that innovation is both a process and a product; change or discontinuity. Innovation cannot be linear; on the contrary, it must change the existing paradigm of how things work [36].

The work of Leurs [37] presented a wide range of innovative approaches that are being used by public-sector managers around the world, without seeking to solve public problems, as is traditionally thought of in the public sector, but rather creating experiments, prototypes, and knowledge about a particular public problem through a process of co-creation with society in order to face the wicked problems.

Among the approaches presented by Leurs [37], we highlight Living Labs, which are between an Intelligence Space ("understanding reality") and a Solution Space ("shaping reality"). ULLs, in turn, can be considered as an expansion of living labs with the objective of supporting innovative solutions, adding social value to society in a process of co-creation through the participation of public and private people who are aiming at social benefits [16], and going beyond the relationship between users and companies. However, since there is not a unique definition of the term ULL, we will delve deeper into this debate in the next section.

### 3.2. Urban Living Labs

As pointed out by Steen and van Bueren [38], even the notion of the ULL has not been clearly defined, as it is treated as a methodology, an environment, a system, or a governance approach. Therefore, in order to investigate the concept of ULLs more deeply, it is necessary to understand the context of the current debate on the topics of smart cities (SCs) and living labs. There is still no consensus among practitioners and academic researchers within the thematic field of smart cities [8]. This concept has been used in various ways under different circumstances, producing several conceptual variants arising from the replacement of the term "intelligent" with other alternative adjectives [39]. The contemporary origins of SCs are related to the "smart growth" movement of the late 1990s in relation to "sustainable urbanization." Most of the initial definitions of the smart city had a strong appeal in the diffusion of ICTs and tended to disregard the importance of other crucial factors outside the scope of technology. More recent approaches included the needs of people and communities, as well as their quality of life, as in the case of the concept of the SC4D (Smart City for Development) derived from ICT4D (Information and Communication Technology for Development) [40]. In this context of collaboration, co-creation, and innovation, living labs emerged. As a result of her research, Bravo Ibarra [11] presented different conceptual definitions of the theme, with two common characteristics: innovation and the participation of the user. The participation of public authorities, although considered important in the literature [41], is not present in most definitions. On the other hand, the idea of innovative processes or solutions with user participation is almost unanimous.

Thereby, following Hossain et al. [42] and Greve et al. [43], this article considers LLs as environments that provide shared resources and bring together multiple stakeholders by using multiple methods of real-life experimentation to create, communicate, and provide new knowledge and validate existing products, services, and processes. The objective is to support innovative solutions in a co-creative process with the participation of public and private people who partner and aim for a common goal, usually creating social benefits.

Although the distinction between the terms living lab and urban living lab is not been agreed upon in the literature, some considerable insights delimit the concepts. An important difference pointed out by Chronéer, Stahlbrost, and Habibipour [44] is that LLs emphasize their actions in the interactions between users and private companies, while ULLs guide their actions toward innovation in solving public problems. In this case, there is an interaction between ULLs and local governments that is focused on creating social value and civic engagement. Another aspect emphasized is the need for a physical location for the innovation of a ULL and the focus on sustainability, especially in the urban dimension of sustainability challenges [14,38]. Considering the different approaches in the literature, in this article, we corroborate the arguments presented by Amorim et al. [16] in a conceptual review of ULLs. They pointed out that these laboratories have the characteristic of a delimited physical space, with emphasis on sustainable urban solutions that require active participation from the local public authorities and a focus on citizens' participation

and validation. We also add the definition by Blezer and Abujidi [20], in which urban living laboratories must be understood as: "( . . . ) a form of experimental governance whereby urban stakeholders develops and test new technologies, products, services and ways of living to produce innovative solutions to the challenges of climate change, resilience, and urban sustainability". This definition seems to us to be quite complete, as it points to the idea of experimental governance [45]. It also leads to the main challenge in cities today: climate change.

Although most academic studies highlighted the expected positive outcomes of experimentation and innovation, some studies expressed some critiques of the implementation of ULLs. Levenda [9] brought us, through a Foucauldian lens of governmentality, some reflections about the dominant motivations for urban experimentation and who the beneficiaries are. Some questions arose on how urban experimentation shapes the approach to sustainability and justice, how experiments engage communities/citizens, and the resulting implications. The author concluded that ULLs take on an exclusionary logic, as the city becomes marketed as a place for large technology companies to test their products and services and as some groups of people are privileged over others. There were some critiques about the elitist bias in the pattern of ULLs being placed in central areas or in gentrified pockets, causing the participation to of citizens be limited to high-class professionals [15]. When ULLs are implemented by companies that pay users for the usage of personal data, usually with tax incentives, the targeted group will most likely be composed of young or lower-income people who already suffer from structural discrimination [46]. The use of "urban data" and research on ethics and privacy concerns were also parts of the observations of Taylor [46] and Veeckman and Temmerman [10]. These criticisms are important to consider when implementing a ULL, especially those related to the use of urban data and to ethical and privacy issues.

In short, public innovation policies for the adaptation of critical infrastructures are urgent. ULLs, in theory, aim to be tools for addressing these issues and enhancing the resilience of the urban environment. To investigate if ULLs really address the critical infrastructure adaptation problem, we will conduct a further content analysis of the case studies presented in the academic literature.

## 4. Results

### 4.1. ULLs and Critical Infrastructure

In this section, a review of the literature on ULLs is presented with a specific focus on the analysis of the case studies presented in the literature in order to verify the projects according to their geographic location and critical infrastructure sector. As this is a recent topic of investigation, there is not a temporal cutoff, and all eligible papers were collected according to our research protocol. The result was, a temporal analysis of the period between 2014 and 2022. This survey will allow the empirical verification of if ULLs focus on sustainability and climate change issues, as suggested by the literature, by using the studies as a proxy.

### 4.2. Bibliometrics of Urban Living Labs

ULLs seek innovative urban solutions to overcome the challenges of climate change, resilience, and sustainability. In this subsection, we present a survey of the main topics addressed in empirical papers on ULLs. There are many methodologies and definitions of what a literature review should be. For the scope of this work, we will use the consolidation used by Cardoso Ermel [19], which differentiated between scientometrics, bibliometrics, and infometrics. In this study, with a focus on the research questions presented in the introduction, we combine aspects of scientometric analyses (to describe the quantitative aspects of a topic) and bibliometric analyses [19] (p. 32). Later, we will conduct part of a content analysis to investigate the origins and characteristics of the case studies presented, always focusing on the geographic locations of the cities and the adherence of the sector to the classification of critical infrastructure explored in the previous section of this paper.

The authors chose to proceed with a research protocol that used Scopus as the primary database (DB), as this DB had a better coverage of all major disciplines [47], and ULLs are a multidisciplinary subject. The search was for the term "urban living lab" OR "urban living labs" in papers published before April 2022; 66 documents were obtained, as described in the following table (Table 2).

**Table 2.** Productions on ULLs.

| Year | # | % | Journals | # | % | Researchers | # | % |
|------|---|---|----------|---|---|-------------|---|---|
| 2022 | 3 | 5% | Sustainability | 14 | 21% | The Netherlands | 21 | 21% |
| 2021 | 25 | 38% | Urban Planning | 8 | 12% | Sweden | 12 | 12% |
| 2020 | 13 | 20% | European Planning Studies | 5 | 8% | Italy | 11 | 11% |
| 2019 | 11 | 17% | Frontiers In Sustainable Cities | 5 | 8% | Germany | 8 | 8% |
| 2018 | 8 | 12% | Journal Of Cleaner Production | 2 | 3% | Australia | 6 | 6% |
| 2017 | 3 | 5% | Solar Energy | 2 | 3% | Belgium | 4 | 4% |
| 2016 | 1 | 2% | Sustainability Science | 2 | 3% | Finland | 4 | 4% |
| 2015 | 1 | 2% | Technology Innovation Management Review | 2 | 3% | Canada | 3 | 3% |
| 2014 | 1 | 2% | Others | 26 | 39% | Others | 33 | 32% |
| | 66 papers | | | 66 papers | | | 102 researchers | |

The findings suggest that the production on the subject is at its peak, since more than 40% of the academic production was published in the period between 2021 and April 2022. The data show that the academic production was heavily concentrated among researchers linked to European institutions—in particular, the Netherlands, Sweden, and Italy. This European dominance will also become evident when analyzing the cases discussed. Finally, in this brief scientometric analysis, we highlight the publications in *Sustainability* (Switzerland), a journal recognized for its plurality and its integrated approach to issues related to the environment, including the urban environment.

A helpful data analysis software used in bibliometric literature reviews [48], VOSviewer, is a tool used to create, view, and explore maps based on network data [49]. This software was used to investigate the texts that were considered as the "core" of the literature in this paper.

Figure 1 shows a visualization of the bibliographic sample. The size of the nodes indicates the number of citations, while the link between these nodes concerns the association strength between them. It is interesting to observe that a considerable number of documents are not related to the main group, which is formed by texts with greater counts of citations and links between them.

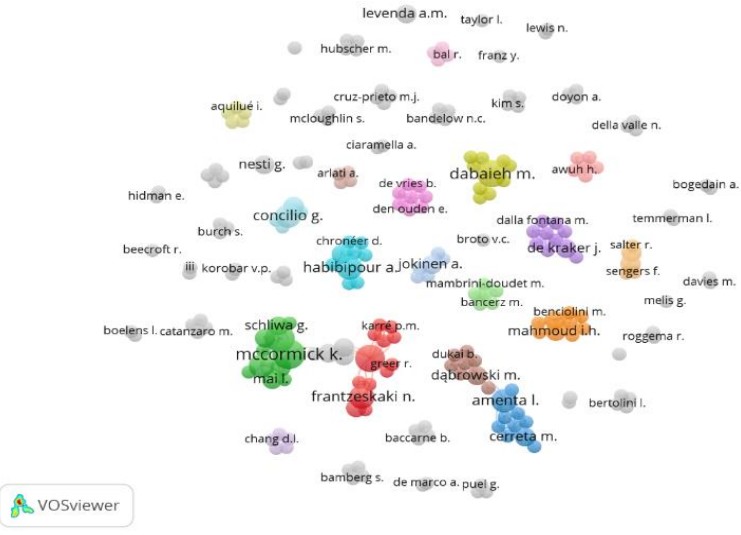

**Figure 1.** Nodes of the bibliographic sample. Source: the authors.

Only five main clusters have some sort of dialogue between them: (i) those led by Kes McCormick [14,50] from Lund University (Sweden) with the ones led by Timo von Wirth [51] of Erasmus University (The Netherlands) and Annica Kronsell [52,53] of Lund University; (ii) those led by Libera Amenta [54,55] from Delft University (The Netherlands) with the one led by Marcin Dąbrowski [56], who was also from Delft University, and Annica Kronsell from Lund University. This lack of integration is very common in themes linked to urban studies. For instance, each area of interest, such as sanitation, transportation, housing, land-use planning, etc., is often ruled by separate departments or spheres, which compete for privileges, resources, or staff [57]. This need for transdisciplinary work is also a common obstacle in the implementation of ULLs [45].

The next step was to analyze, through practical screening techniques [18,19], all of the papers focusing on the following question: When case studies were presented, which countries hosted more interventions and in what sector were they related to the critical infrastructure of cities? The results are summarized in the following table (Table 3).

**Table 3.** Countries and cities with ULL interventions.

| Country/City | | # of ULLs | % Total | # of Papers Cited | % Total |
|---|---|---|---|---|---|
| **The Netherlands** | | 25 | | 34 | |
| | Amsterdam | 10 | 25% | 13 | 28% |
| | Rotterdam | 10 | | 15 | |
| | Others | 5 | | 6 | |
| **Sweden** | | 10 | | 12 | |
| | Lund | 3 | 10% | 3 | 10% |
| | Malmo | 3 | | 5 | |
| | Others | 4 | | 4 | |
| **Germany** | | 10 | | 11 | |
| | Hamburg | 2 | 10% | 3 | 9% |
| | Others | 8 | | 8 | |
| **Finland** | | 8 | | 10 | |
| | Tampere | 5 | 8% | 7 | 8% |
| | Others | 3 | | 3 | |
| **Other European countries** (Austria, Belgium, Czech Republic, Denmark, France, Hungary, Italy, Kosovo, Macedonia, Norway, Poland, Portugal, Spain, UK) | | 33 | 33% | 38 | 31% |
| **Oceania** (Australia) | | 5 | 5% | 5 | 4% |
| **Asia** (Hong Kong, Japan, Qatar, Turkey) | | 4 | 4% | 5 | 4% |
| **Americas** (USA, Mexico) | | 4 | 4% | 4 | 3% |
| **Africa** (Egypt) | | 2 | 2% | 2 | 2% |
| **Total** | | 101 | 100% | 121 | 100% |
| **Global South/Low income** | | 5 | 5% | 6 | 5% |
| **Global North/High income** | | 96 | 95% | 115 | 95% |

We focused on two main items: the numbers of ULLs presented as part of the case studies and occurrence in many different studies. Once again, the data show that ULLs can be characterized as a European phenomenon, with the Netherlands leading the way. Only the cities of Amsterdam and Rotterdam had more ULL experiences than in any other country. These results will guide the discussion about the transfer of knowledge from developed countries to the Global South, since only 5% of the cases were from low-income cities, and only one of them was from a Latin American city.

Finally, the most subjective part of the content analysis was the classification of ULLs according to the main type of critical infrastructure to be adapted/improved. Many of the laboratories did not have a single sectoral orientation. A critical analysis of relevance was chosen by defining up to two sectors per laboratory. Using the definitions presented

in the previous section, the main challenge was the division between physical and green infrastructures (or nature-based solutions). One example was energy; despite being an iconic sector of traditional critical infrastructure, the projects analyzed were more linked to energy efficiency and off-grid solutions. With this perspective, we chose to classify them as green. The only sector that seemed to us to be more deeply linked to traditional physical structures was urban mobility, since even solutions that encourage the use of bicycles, for example, end up using streets and roads, which are jointly considered as physical infrastructure sectors (Table 4).

**Table 4.** Critical infrastructure sectors of ULLs.

| ULL Sector | CI Type | % | N |
|---|---|---|---|
| Citizen Participation Urban Governance Urbanism | Culture | 40% | 46 |
| Energy Housing Nature-Based Solutions (Drainage) Waste Water | Green or nature-based | 40% | 46 |
| Circular Economy Health Urban Food Policy Academic Research Innovation | Institutional | 11% | 13 |
| Urban Mobility | Physical | 9% | 10 |

The main conclusion is that most ULLs focus on socio-cultural and institutional aspects within the critical infrastructure classifications—in particular, citizen participation, urban design, and circular economy mechanisms. When approaching the experiences of sectors that demand more financial resources, the emphasis is on experiments that focus on innovative and nature-based aspects of traditional sectors, such as energy, waste, and water.

It should be noted that many projects are transnational, and the majority are financed by government institutions, such as the European Union. As examples, we can mention the FURNISH [58] (Fast Urban Responses for New Inclusive Spaces and Habitat), M-NEX [59] (design for the food–energy–water nexus), and UnaLab [3,44,60] (Urban Nature Labs) projects. These projects have the characteristics of knowledge exchange between cities, and all of them are in the Global North.

*4.3. Knowledge Transfer of ULLs*

Sharing knowledge between laboratories is one of the pillars of the open innovation process. However, this exchange mechanism is still the cause of many difficulties between various local teams. Sarabi and colleagues [45] discussed the difficulties in establishing innovative governance for the implementation of ULLs focused on nature-based solutions in three European cities (Tampere, Finland; Eindhoven, The Netherlands; Genoa, Italy). Dąbrowski, Varjú, and Amenta [56] also investigated this knowledge sharing between two ULLs from Amsterdam, The Netherlands and Naples, Italy. Both studies had similar conclusions: Transfer of knowledge policies or "best practices" has become a usual showcase of policymaking for contemporary urban planning. In addition, just "copying and pasting" may lead to suboptimal solutions if one does not consider the disciplinary, geographic, socio-cultural, and governance/decision-making background [56]. The following table (Table 5) summarizes the main barriers to this knowledge exchange based on the work of Sarabi and colleagues [45].

**Table 5.** Barriers for ULL adoption for NBS.

| Organizational and Structural Barriers | Cognitive, Cultural, and Behavioral Barriers | Knowledge and Process Barriers | Ethical Barriers |
| --- | --- | --- | --- |
| Lack of political will and long-term commitment | Negative past experiences | Uncertainties regarding the added value and benefits of ULLs | Intellectual property (IP) |
| Lack of supportive legal and policy frameworks | Perceived complexity of the ULL approach | Lack of available guidelines and tools for engagement | Privacy issues |
| Disconnection from the mainstream development process | Risk aversion and reluctance to change | NBS monitoring and assessment challenges | Inclusiveness |
| Sectoral silos | Conflicting expectations | Lack of skilled knowledge brokers | - |
| Inflexible hierarchical organizational structure | Lack of public awareness and engagement | Inability to upscale and replicate projects | - |
| Lack of sufficient human resources | Lack of engagement to take responsibility | Lack of learning from other experiences | - |
| Lack of sustainable financial resources | - | - | - |

However, it is important to understand the barriers—especially the political and institutional ones—as well as the interdependence that exists in order to overcome them and establish a ULL. One of the patterns is that of difficulties with the local government and other stakeholders. This aspect was also highlighted in the works of Kronsell and Mukhtar-Landgren [53] and Oldenhof et al. [15], who pointed out that the proximity between a lab and the local government causes municipalities to inhibit actively innovative processes as a result of local political priorities or, in an indirect sense, with institutional obstacles that make processes more difficult and, thus, create barriers to management innovations.

The need for leadership, ownership, and management stresses the delicate balance between steering or controlling and a lab's need for flexibility and effectiveness [44]. Management factors (legitimacy, responsiveness, stable funding, leadership) and contextual factors (path dependency, political environment, demographics, good governance) are also crucial in determining how effective contributions from partners are [15]. Stakeholder engagement is usually a complicated and messy process that is tainted with conflict, disagreement, and diverging points of view [3]. According to Klautzer and colleagues [13], involving citizens and other stakeholders in the decision-making processes concerning the purpose, design, and construction materials and techniques will foster knowledge sharing. Finally, the availability of external financial sources in the long run and lack of pollical will seem to be a great issue for ULLs in terms of lasting beyond the period and scope of an individual project's budget [45].

This problem is worsened when seen through a critical analysis of the potential gains and constraints of ULLs in cities in developing countries. The main scope of a ULL is to offer an open and collaborative environment that considers citizens as agents in urban transformation processes and enables the exchange and co-creation of shared value in a city [16]. However, the contexts and challenges of cities in developing countries, which present a combination of social vulnerability and climate change, are quite different from those in the European context. In this sense, analyzing the performance of these laboratories in the process of producing public policies in an adverse context becomes fundamental to understanding how this new tool can help cities to mitigate real problems that are being caused by climate change.

National and international institutions usually develop urban policies that provide homogeneous solutions to heterogeneous problems, such as housing [61]. This type of conflict, which is very common in public organizations, is linked to the concept of mimetic isomorphism, where a copy of solutions and best practices is recommended and the fo-

cus is on gains in legitimacy rather than functionality [62]. However, complex problems do not have a "golden rule", especially when there are large numbers of human transactions, autonomous street agents, and different stakeholders in a dynamic network of interactions [63]. This seems to be the case for urban living labs.

The challenges of isomorphism when implementing urban public policies using tools from developed countries (Global North), such as urban living labs, in cities in the Global South have been extensively studied in the literature on urban governance [64]. A simple transfer of the design and implementation of public policies and practices without taking into account the needs, capacities, and political and institutional contexts of emerging countries can lead to so-called "Type-Three Errors" (i.e., solving the wrong problem) or even the emergence of new and more complex issues [65] (p. 14). The idea that low-income countries may have a so-called "tyranny of emergency" [64], which can lead to a low emphasis on the planning aspect of urban policies and the rise of the everyday informalities [66], cannot be disregarded.

When working with innovation policies, this aspect is amplified, since such policies are inspired by models in which there is an abundance of human and financial capital [67]. Kuhlmann and Ordóñez-Matamoros [65] argued that, in order to have political and social legitimacy, where innovation is not seen only as a "mantra" of the elite, one must adopt a progressive vision of innovation. Local aspects, such as creativity, resourcefulness, and knowledge of local and marginalized peoples, need to be made appropriate with the support of the government [65].

Of the 66 papers found in the Scopus database, few had sections on "best practices and recommendations", and all of them were focused on European ULLs. Tanda and De Marco [68], Veeckman and Temmerman [10], and Arlati et al. [69] highlighted the aspect of the engagement of local communities with the action of feedback from both utilizers and citizens, especially with disadvantaged social groups who play a direct role in the implementation. The role of public actors in the design process and the streamlining of public procurement processes were also common recommendations. As shown previously, only four studies analyzed case studies in low-income and/or Global South countries: Egypt [70,71], with housing energy efficiency projects; Mexico [72], in an efficient building within a university; Macedonia [73], using communication techniques to increase bicycle use for mobility; Kosovo [74], using urban data sensors to support collaborative urban planning. These projects were specific and limited, with a narrow scope and external funding. They did not seem to be part of a long-term solution in order to tackle climate change issues or to offer a major contribution to the adaptation of critical infrastructure.

In the case of ULLs, social innovation and effective and broad collaboration are clear principles for their characterization as a locus of co-creation of solutions that bring public value [38], notwithstanding that part of the literature sees collaboration, especially in collective projects, as a "value" or superior principle to be followed. It should be noted that every process of participation in social innovation has significant costs, and many consider it necessary to carry out an analysis of the costs and benefits of the open governance process, stating that "collaboration must be critically evaluated and not sought for itself" [75].

Governance failures are common in innovation processes in emerging countries and are one of the factors that most explain, at least in part, why knowledge and innovation fail to successfully contribute to the progress of these countries [65]. All of these considerations must be observed when implementing a ULL in a Latin American country, for example. Of course, citizen participation is important for open knowledge, but it is important to highlight the importance of an open and collaborative process being evaluated before being used as a public policy tool. This reflection is significant, considering that most of the literature always deals with innovation and its tools from a positive perspective. However, we need to consider the local contexts to assess whether a given tool will, in fact, be useful for the implementation of a policy or for the innovation of a process. In the next section, the Brazilian experience with ULLs will be presented while applying the concepts and reflections shown so far.

*4.4. The Brazilian Experience*

In this section, we present the Brazilian experience with ULLs with the objective of understanding the logic of the implementation of these laboratories in Brazil. The following figure (Figure 2) presents a summary with the main keywords of the objectives of the ULLs collected in our sample. We can observe words that are key in the concept of ULLs used in this paper, such as social, development, environmental, solutions, innovations, cities, and urban, among others.

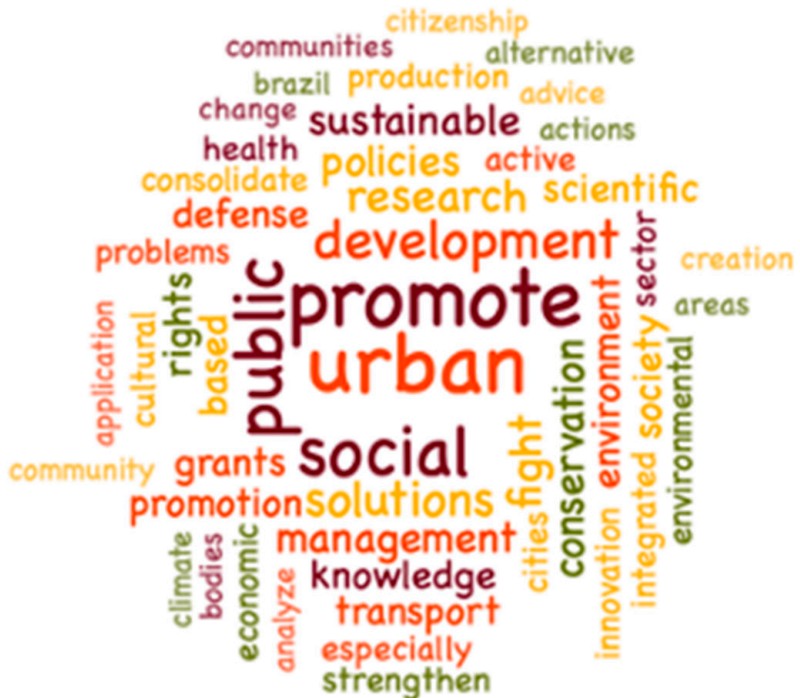

**Figure 2.** Word clouds of ULL goals (selected sample). Source: the authors.

This corroborates the validity of the criteria used in this research for the selection of laboratories. This can be confirmed in the following quotes about the objectives of some of the ULLs.

"Supporting cities to achieve healthy food systems for people and the planet, resilient to climate and economic vulnerabilities, and promoting social justice, based on the democratic construction of integrated and coherent policies that address urban food challenges in a systemic way" (LUPPA) [76].

"The central scope of the Center for Efficiency in Urban Sustainability–CESU/Teresina is to validate, in the urban environment, Climate Technologies that promote the reduction of carbon emissions and provide a better quality of life for the population of the City. To this end, the Center will promote the validation of climate technologies that will support carbon emission reduction and the 4.0 economy. CESU/Teresina is a ULL aimed at validating solutions to urban public problems, with a focus on mitigating the effects of climate change" (CESU/Teresina) [77].

"(...)Promote education for the strengthening of democracy through actions that encourage social engagement and the exercise of citizenship, through the participation of society in the formulation of public policies. Promote and consolidate the concept of sustainable city in public and private university (...)" (Laboratório da Cidade) [78].

Blezer and Abujidi [20] (p. 76) presented a typology with three models of ULLs based on the variables of geographic scale and urban dimension. The first is the strategic one, which encompasses large-scale technological development programs that involve public and private actors. The objective is to increase a city's ability to compete internationally. The second is the *civic* model, which tends to limit its involvement to local governments

and local actors. The scope is focused on local priorities. Lastly, there is the *organic* model, which specifically focuses on local issues related to urban poverty. The key actors are organized civil society, not the state or the private sector.

Table 6 presents our sample of 13 Brazilian ULLs. Ten can be classified as civic ULLs, as they interact with the state (local or national) and the private sector. This is reflected in the funding of the laboratories, as the vast majority depend on public resources. The origin of the resources varies—whether local governments, the federal government, universities, or state research. There are some laboratories that, in addition to government funding, also receive funds from multilateral organizations. Of all of the cases, only one has private funding.

The funding of laboratories is a relevant topic that needs to be explored in greater depth by academia, since there is strong criticism in the context of the Global North of the loss of autonomy of cities due to big technology companies that offer technological services in exchange for "only" data of its citizens [79]. In this sense, ULLs in Brazil, at least those collected in our sample, are mostly *organic* and funded by the state, not by companies interested in product validation and citizen data. In addition, in the context of critical infrastructure, all of them are focused on cultural/institutional types, with some actions toward green or nature-based solutions.

Considering that public funding in developing countries is scarce, this model can limit the expansion of ULLs. In this case, the questions that can be addressed to this topic are: *How can private companies be brought to participate in the ULLs in Brazil?* We do not have an answer to this question, and we understand that this problem is related to the broader objective of a ULL, which is to add social value while focusing on public problems.

Regarding governance, most ULLs have a collegiate decision-making process, demonstrating a horizontality in the way that these laboratories work, as expected due to the literature [53]. However, the main governance challenge does not refer to the internal relationships of the actors that make up a ULL, but the relationship with the stakeholders: the municipality, private-sector actors, and citizens [45]. In the Brazilian case, we can add one more actor: the federal government. In our research, of the 13 ULLs, four are funded by federal institutions, which increases the cost of negotiation in order to reach a consensus on a given decision. In Brazil, municipalities have administrative and financial autonomy. However, in practice, the national government concentrates most of the public resources and transfers them to states and municipalities. Concerning gender diversity, eight of them have 50% or more of women in their composition, which indicates that they are open spaces and that there is no male predominance. The strong presence of women may be related to the informality of some ULLs, given that, in Brazil, women still have difficulty standing out in formal environments, especially in leadership roles. In any case, we consider gender diversity a positive aspect in the investigated laboratories.

The last aspect to highlight is the emergence of this tool in the process of producing public policies in Brazil. The two oldest labs in our sample were created in 2014, just 8 years ago. In this sense, this is still a topic that deserves much attention from academia in terms of its developments in cities and in the field of public policy.

Although the themes of sustainability and climate change are primary or important for most of the 13 ULLs analyzed in Brazil, we were not able to identify an action with a significant impact on critical infrastructure that tackles the climate change challenge. Thus, we cannot refer to the implementation of ULLs in Brazil as a national strategy for pursing resilience to the effects of climate change.

**Table 6.** Brazilian ULLs.

| Lab | # | Start Year | Decision Governance | % Women | Interaction with the Public Sector | Funding | Legal Structure | Economy Sector |
|---|---|---|---|---|---|---|---|---|
| LUV | 6 | 2019 | No data | 50 | No data | No data | Social Organization | Third Sector |
| Laboratório de Construções Inteligentes—LCI/Coppe UFRJ | 21 | 2019 | No data | 38 | Yes | Federal Government | Research Center | Academy |
| TransLabUrb | 10 | 2014 | Collegiate | 25 | Yes | Local Governments and Multilateral development bank | Social Organization | Third Sector |
| ODSLab | 15 | 2017 | No data | 40 | Yes | NGOs and think tanks, European Union | Social Organization | Third Sector |
| GNOVA | 10 | 2016 | Collegiate | 90 | Yes | Federal Government | Public Sector | Public Sector |
| Laboratório Urbano de Políticas Públicas Alimentares —LUPPA | 5 | 2021 | Monocratic | 100 | Yes | NGO S | Social Organization | Third Sector |
| Sociedade Global—Laboratório de Inovação Urbana | 9 | 2019 | No data | 67 | Yes | no data | Social Organization | Third Sector |
| Laboratório da Cidade | 19 | 2018 | Collegiate | 67 | Yes | NGOs, Government | Social Organization | Third Sector |
| Novos Urbanos | No data | 2015 | No data | | Yes | Private Companies; Think tanks | Social Organization | Third Sector |
| Lab Pro comum | 16 | 2017 | Collegiate | 62 | No | NGO | Social Organization | Third Sector |
| Lab Jaca | 11 | 2020 | Collegiate | 54 | No | Federal Government, Regional government | People's Initiative | Third Sector |
| MobiLAB | 3 | 2014 | Collegiate | 33 | Yes | Regional Government | Research Center | Academy |
| CESU-Teresina | 9 | 2021 | Collegiate | 67 | Yes | Federal Government | Research Center | Academy |



## 5. Discussion and Conclusions

The discussions carried out throughout this article corroborate the idea that the ULL is a new tool for city management focused on the theme of urban sustainability and the co-participation of citizens in the solution of public problems. Although there are criticisms in the literature that point out barriers to the implementation of effective citizen co-participation, as well as problems related to ethical issues and the use of data from cities, we understand that ULLs are new instruments of governance that, even in Europe, are still in a construction process.

From the typology of critical infrastructure presented in Table 1, the innovation of public policies aimed at mitigating the effects of climate change should especially focus on physical and green initiatives, as they generate greater impacts on the mitigation of climate effects. Although often associated with co-creation and the implementation of sustainable solutions, in practice and according to the data collected from the systematic collection of case studies presented in the literature, most projects are associated with low-budget interventions, with an emphasis on building a culture of sustainability and stakeholder engagement. In relation to physical infrastructure, which demands large and long-term investment projects, little attention was dedicated by ULLs to the enhancement of safer, more resilient, and more sustainable design strategies and frameworks.

The population living in cities in the Global South is the most vulnerable to climate change, and the infrastructure needs to mitigate the effects and bring dignity, which depends on a massive amount of financial resources. New ways of providing these infrastructures are needed, demanding innovative management processes and technologies. Few ULLs are being operationalized in the Global South and, as we have seen in the Brazilian case, they are more related to civic engagement actions, with few deliveries in terms of infrastructural innovation, which is probably due to the financial limitations of the labs. In summary, ULLs—even in Europe—are quite limited instruments in terms of achieving the goals of the adaptation of critical infrastructure in the face of environmental problems.

So, would an urban living lab be a utopia for large-scale solutions to global problems, especially in the periphery of Global South countries? In a certain way, yes. It is not possible to expect that technological solutions or the creation of these environments will be a definitive solution that will mitigate the effects of climate change. However, we claim that the role of knowledge co-creation and co-production in these laboratories is of inestimable value for the awareness of citizens and stakeholders, as it directs public resources and attention to the adaptation of critical infrastructures.

*Limitations and Research Agenda*

This study has some limitations. Although the sample used here was quite extensive, it did not encompass all publications on urban living labs, especially documents that investigated similar interventions that had different names (e.g., city labs, urban labs, change labs, urban transition labs, sustainable living labs, smart city initiatives). The Brazilian sample and investigation may demand a deeper analysis with more in situ case studies.

We stress the need to recognize that the subject of ULLs deserves attention from both academics and institutions that aim to debate the future of cities and the impacts of climate change (such as governments, think tanks, banks, and private companies). Although the growing number of publications on the subject is recognized, there are still many gaps to be addressed in future studies. Hence, the following research avenues are suggested:

- In terms of a partnership model, more studies are needed about the economic value generated by companies, their business models, and the feasibility of interventions in the long run.
- Can we understand the ULL as a "democratic experiment" that promotes more sustainable development in cities?
- What are the state capacities needed to implement a ULL? How does the local government's capacity influence the success of a ULL in cities of the Global North and South?



- Flexibility in the legal framework is key for the successful implementation of a ULL as a platform for urban innovation. Further studies on how different legal arrangements can hinder or boost a ULL are recommended.
- Most ULLs were financed by public funds. What is the actual cost/benefit of these experimentations? How can it be measured?
- Are the timing of the implementation and the outcomes of a ULL compatible with the urge for sustainable urban solutions in order to achieve global sustainability agendas, such as the Paris Agreement?
- How does the bias of the education level of participants influence the design and the acceptance of the solutions?
- With this variety of definitions, institutional or organizational designs, and actors, how can one structure a minimum-governance structure that allows urban innovation to become a continued project beyond the laboratory?

Finally, despite being a promising approach to the most pressing urban issues, it is essential to study whether the results can be possibly replicated in a large variety of contexts around the world, whether they will have any significant effect on the creation of effective adaptation and mitigation strategies, and whether they will contribute to an intelligent city (society) that is, after all, human and inclusive.

**Author Contributions:** Conceptualization, E.E.R.A. and M.M.; methodology, E.E.R.A. and M.M; software, E.E.R.A. and M.M.; validation, E.E.R.A., M.M. and K.V.G.F.; formal analysis, E.E.R.A. and M.M.; investigation, E.E.R.A., M.M. and K.V.G.F.; resources, E.E.R.A., M.M. and K.V.G.F.; data curation, E.E.R.A., M.M. and K.V.G.F.; writing—original draft preparation, E.E.R.A., M.M. and K.V.G.F.; writing—review and editing, E.E.R.A., M.M. and K.V.G.F.; visualization, E.E.R.A. and M.M.; supervision, E.E.R.A. and M.M.; project administration, E.E.R.A. and M.M.; funding acquisition, M.M. All authors have read and agreed to the published version of the manuscript.

**Funding:** This research was part of the Project "Centro de Eficiência em Sustentabilidade urbana—CESU Teresina" and was funded through the Conselho Nacional de Desenvolvimento Científico e Tecnológico—CNPQ (404290/2020-5).

**Institutional Review Board Statement:** Not applicable.

**Informed Consent Statement:** Not applicable.

**Data Availability Statement:** The data presented in this study are available on request from the corresponding author.

**Acknowledgments:** We would like to thank the collaborators and interns of the CESU team.

**Conflicts of Interest:** The authors declare no conflict of interest.

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
