# Peer review of "Urban Living Labs and Critical Infrastructure Resilience: A Global Match?"

_sustainability, doi:10.3390/su14169826_

Round 1
Reviewer 1 Report
Thank you to the authors for making the submission. This is an interesting paper focusing on Urban Living Labs and Critical Infrastructure in cities. The manuscript has potential to be published in the journal provided the following comments are considered/addressed to strengthen it further.
I would suggest reviewing the title of the paper in line with the main body.
A gap in the literature needs to be clearly highlighted which may inform the current study (research question). This should result from the critical literature review. This gap should inform the current study and research question.
Original contribution to knowledge needs to be strengthened in the manuscript.
All the methodological decisions and choices need to be appropriately justified in the context of the current research. Why were these qualitative and quantitative data collection methods chosen? Discuss all the practical details for clarity and transparency.
I am not sure if ‘Materials and Methods’ should be before ‘Definitions and Conceptualization’ section?. Review the structure of the manuscript for better flow and alignment.
The case study (4.3. The Brazilian Experience) analysis needs some more in-depth discussion. It feels descriptive at present and needs to offer some more critical insights in line with the research question/s.
The manuscript needs to have some recommendations for practitioners and policymakers in cities arising from the study findings.
Writing the manuscript in first person can be avoided, for example, using ‘we’. Check the journal guidelines for this purpose though.
The manuscript needs to be proofread at the end to address any language related errors/typos.
Author Response
- I would suggest reviewing the title of the paper in line with the main body.
Response: Suggestion accepted. Our primary focus is critical infrastructure resilience due to climate change in the Global South region
- A gap in the literature needs to be clearly highlighted which may inform the current study (research question). This should result from the critical literature review. This gap should inform the current study and research question. Original contribution to knowledge needs to be strengthened in the manuscript.
Response : Thank you for the observation. Suggestion accepted. We made our research questions and literature gap contributions more explicit.
All the methodological decisions and choices need to be appropriately justified in the context of the current research. Why were these qualitative and quantitative data collection methods chosen? Discuss all the practical details for clarity and transparency.
Response : Thank you for the observation. Suggestion accepted .We made it more explicit the methods used and the choice for scopus
- I am not sure if ‘Materials and Methods’ should be before ‘Definitions and Conceptualization’ section?. Review the structure of the manuscript for better flow and alignment.
Response : We appreciated the suggestion, but after a careful review of other papers in this journal we saw no patterns of the section Materials and Methods. There are many examples like:
Keller J, Scagnetti C, Albrecht S. The Relevance of Recyclability for the Life Cycle Assessment of Packaging Based on Design for Life Cycle. Sustainability. 2022; 14(7):4076. https://doi.org/10.3390/su14074076
Takyia, Harmi, at. al Application of Open Government Data to Sustainable City Indicators: A Megacity Case Study. Sustainability 2022, 14, 8802. https://doi.org/10.3390/su14148802
Therefore, we prefer to maintain the original order with the understanding that the flow of information is adequately understood.
- The case study (4.3. The Brazilian Experience) analysis needs some more in-depth discussion. It feels descriptive at present and needs to offer some more critical insights in line with the research question/s.
Response: Thank you for the observation. Indeed, there is no much data available for a more in-depth discussion. We highlighted the preliminary and descriptive aspects of the conclusions
- The manuscript needs to have some recommendations for practitioners and policymakers in cities arising from the study findings.
Response: Suggestion accepted. We made it more explicit what our research could contribute to practitioners and policymakers, especially those in the global south.
- Writing the manuscript in first person can be avoided, for example, using ‘we’. Check the journal guidelines for this purpose though.
Response 1: Suggestion accepted. As seen on the file attached, we tried rephrasing many of the sentences originally in first person.
- The manuscript needs to be proofread at the end to address any language related errors/typos.
Response 1: : Suggestion accepted. As seen on the attached file, we sent it to an English native researcher that contributed to the correctness of the writing.
Reviewer 2 Report
Dear Authors,
Please find my comments about the manuscript: Urban Living Labs and Critical Infrastructure: a global match?
The manuscript falls within the scope of SUSTAINABILITY, and in my opinion, it is of great interest and represents a valuable approach to understand the role of urban living labs (ULL) to promote a more sustainable urban future.
The paper represents a broad review of previous work and is a very valuable contribution to understanding the complexity of urban planning.
Despite the case study was carried out in Brazil, the results could be applied to different situations including countries of Global South with low income.
Minor suggestions could be done to the authors. One of them is why is not included the United Nations Sustainable Development Goals in the analysis.
Author Response
- One of them is why is not included the United Nations Sustainable Development Goals in the analysis.
Response: A good observation. Although We believe that ULL can have applications in many SDGs, we made it explicited the link on SDG 11 “Making cities and human settlements including, safe, resilient and sustainable.”